# Parity Breaking in Ferrofluids with Vorticity-Magnetization Coupling

Dylan Reynolds, Gustavo M. Monteiro,* and Sriram Ganeshan

*Department of Physics, City College, City University of New York, New York, NY 10031, USA and*
*CUNY Graduate Center, New York, NY 10031*
(Dated: July 18, 2024)

Ferrofluids are synthetic magnetic colloids consisting of magnetized nanoparticles surrounded by a repulsive surfactant layer. When subjected to an external magnetic field the ferrofluid acquires a macroscopic magnetization density which leads to magnetic behavior that is intricately coupled to the ambient fluid dynamics. Ferrofluids share several features with the chiral active fluids composed of unidirectionally spinning hematite cubes, which have been shown to possess a 2D non-dissipative odd viscosity term [1]. In standard ferrofluid dynamics, 3D versions of parity-breaking terms are not commonly observed, partly because of the small size of the magnetic particles. In this work, we investigate if there are unique mechanisms in ferrofluids that can lead to a 3D odd viscosity term. Our results show that coupling the fluid vorticity ($\vec{\omega}$) to the magnetization ($\vec{M}$) with a term proportional to $\vec{\omega} \cdot \vec{M}$ leads to parity breaking in ferrofluid hydrodynamics, and results in a three-dimensional odd viscosity term when the magnetization is relaxed along the direction of a uniform and static applied field. Hele-Shaw cells are commonly used devices to investigate ferrofluids and we demonstrate that this coupling reproduces the parity odd generalization of Darcy's Law discussed in a recent work [2]. A potential experimental setup is discussed which may reveal the presence of this coupling in a ferrofluid confined to a Hele-Shaw cell.

## I. INTRODUCTION

Parity violation is ubiquitous in physical systems, from astronomical and geological phenomena at the largest of scales [3–5], to the superfluid and quantum hall properties of electrons in condensed matter systems [6–22]. In describing the collective dynamics of many particle systems, parity breaking effects typically originate from the presence of intrinsic angular momentum at the level of the constituent particles. Recently much theoretical work has focused on describing so-called "active matter", systems in which energy is not conserved at the microscopic level. These systems naturally exhibit parity-breaking (parity odd) behavior [23–25]. Parity odd behavior in active matter systems can be described in a variety of ways. Odd elasticity is a framework that aims to capture these effects by studying the relation between stresses and strains for non-conservative microscopic interactions [26], while odd ideal gas descriptions study chiral collisions within a kinetic theory framework [27]. Additionally, many chiral active systems are best described using a fluid description [1, 28].

In contrast to ordinary incompressible fluids, these active fluids manifest additional transport coefficients beyond shear viscosity. Perhaps the most famous of them is the rotational viscosity, which acts as a relaxation mechanism for the fluid particle angular momenta. In 2D, there is an extra viscosity known as "odd" or "Hall" viscosity, which preserves the fluid isotropy. Different from the other viscosity coefficients, odd viscosity is neither dissipative nor invariant under parity symmetry. Furthermore, its effects are only present for particular boundary conditions, such as free surface problems [29–31]. On the other hand, parity breaking in 3D is incompatible with isotropy, leading to a much larger class of transport coefficients [32]. However, we recently showed that within a Hele-Shaw cell, the 3D parity breaking effects in incompressible fluids are revealed in a strikingly simple fashion [2, 32].

In this work, we investigate one particular 3D incompressible fluid that appears to possess several characteristics associated with parity-odd behavior, namely, ferrofluids. Briefly, ferrofluids consist of magnetic nanoparticles coated in a hydrophobic surfactant layer which are suspended in a carrier fluid. Each nanoparticle has an intrinsic magnetic moment, which is the sum of the magnetic moments generated by the atoms within it. Without an external magnetic field, thermal fluctuations prevent the alignment of these moments, and the fluid is non-magnetic. When a substantial magnetic field is applied the nanoparticles show collective behavior and the fluid becomes magnetized. For a detailed description of ferrofluids see [33–39]. Two common laboratory setups are that of a steady applied field, where the magnetization density $\vec{M}$ aligns with the external field [40, 41], and a rotating applied field, in which $\vec{M}$ rotates with the external field with the same angular frequency but with a phase lag [42–44].

The case of a rotating applied field, typically termed 'spin-up' flow, in principle should result in parity odd behavior as the rotating field causes the magnetic particles to rotate, which introduces intrinsic angular momentum into the system at the microscopic level. However, in the limit of the vanishing moment of inertia per particle, due to its small size ($I \sim r^2 \to 0$), the microscopic angular momentum does not manifest itself at the hydrodynamic scale. Thus, there is no parity-breaking behavior for fer-

---

* Present address: Department of Physics and Astronomy, College of Staten Island, City University of New York, New York, NY 10314, USA

rofluids in this setup. This should be contrasted to the configuration investigated in the recent chiral active colloids of Soni et al [1], where the magnetic colloids, which are orders of magnitude larger than ferrofluid nanoparticles, show experimental signatures of parity odd effects in the form of 2D odd viscosity. Furthermore, the case of steady applied field trivially does not lead to any parity odd transport coefficients, since no net angular momentum is generated at the macroscopic level.

Thus, it seems that in order for ferrofluids to manifest a 3D analog of odd viscosity, one needs the magnetization to play the role of the fluid intrinsic angular momentum, since the latter usually gets washed out due to the small size of the ferrofluid particles. Following this intuition, in this paper we propose a coupling between magnetization density $\vec{M}$ and fluid vorticity $\vec{\omega}$, which after relaxing the direction of magnetization leads to the 3D odd viscosity in the case of a steady applied field. This coupling is motivated by the recent work of Markovich and Lubensky [24], where they have considered the intrinsic angular momentum coupling to fluid vorticity using a Poisson bracket formalism. The key feature of the vorticity-magnetization coupling is that the appearance of parity odd behavior at the macroscopic scales does not necessarily result from the angular momentum of the ferrofluid nanoparticles. This behavior should be present even in the limit of vanishing particle angular momentum.

The equations of motion for ferrohydrodynamics are usually derived from a detailed analysis of the microscopic forces and torques acting on the fluid [33]. However, to understand the origins of the magnetization-vorticity coupling it's more insightful to use a Poisson bracket (PB) formalism [45, 46], where the dissipative terms are incorporated through a dissipation function. Within this setup, the relevant three-dimensional parity breaking arises when we add the following *intrinsic* term to the ferrohydrodynamic Hamiltonian

$$H_{M\omega} = \frac{\gamma}{2} \int d^3r \, \vec{\omega} \cdot \vec{M}, \qquad (1)$$

where $\vec{\omega}$ is the fluid vorticity, $\vec{M}$ is the magnetization per particle and $\gamma$ is a coupling constant that depends on the microscopic properties of the ferrofluid nanoparticles. As pointed out before, this term is analogous in form to that seen in Markovich and Lubensky [24], with the intrinsic angular momentum $\vec{\ell}$ replaced by $\gamma \vec{M}$. In that analysis it was seen that such term leads to what the authors referred to as 3D odd viscosity. This 3D odd viscosity breaks the fluid isotropy, hence, the corresponding viscosity term contains a transverse part (with respect to $\vec{M}$), which has the same form as the 2D odd viscosity, and a longitudinal part, which is only present in three dimensional systems.

When this term is included in a ferrofluid system it can lead to a variety of parity-breaking behaviors. However, this paper will focus specifically on the case of a constant external field, and when the ferrofluid is confined to a Hele-Shaw cell. In this scenario the system is described by a modified version of Darcy's law, which was previously derived for general parity-odd flows in Ref. 2. Using a Hele-Shaw cell provides a reliable experimental setup to measure the strength of this new coupling.

This paper is organized as follows. In section II we review the standard equations describing ferrofluids and show how they can arise from a Hamiltonian and PB structure. In section III we introduce a vorticity-magnetization term into the ferrofluid Hamiltonian and give the modified equations of motion. In section IV we review some aspects of the anisotropic viscosity tensor in 3D, and in section V we confine the ferrofluid to a Hele-Shaw cell to examine the modification to Darcy's Law.

## II. REVIEW OF STANDARD FERROFLUID DYNAMICS

To understand the significance of our modification, this section will provide an overview of the standard ferrofluid system and its governing equations. The ferrofluid nanoparticles are assumed to be evenly distributed throughout the carrier fluid, and the effects of the surfactant layer surrounding each particle are typically not considered [47]. The ferrofluid is treated as a single entity with a constant density $\rho$, velocity $v_i$, pressure $P$, particle angular velocity $\Omega_i$, and magnetization density $M_i$. The applied field is denoted by $B_i$, and throughout this paper, we will employ Einstein summation notation on repeated indices.

The fluid is assumed to be incompressible with good approximation, $\partial_i v_i = 0$, and satisfy

$$D_t v_i + \partial_i P = \nu \nabla^2 v_i + M_j \partial_j B_i - \Gamma \epsilon_{ijk} \partial_j \left( \omega_k - 2\Omega_k \right), \qquad (2)$$

where the material derivative is defined as $D_t = \partial_t + v_j \partial_j$, $\epsilon_{ijk}$ is the Levi-Civita symbol, $\nu$ is the kinematic shear viscosity, and $\Gamma$ is the kinematic rotational viscosity, which creates a drag force whenever the local particle rotation differs from (half) the local vorticity $\omega_i$. Note that in the above equation, and throughout this paper, the pressure $P$ has been scaled by the density, which has subsequently been set to unity. Conservation of local angular momentum gives the equation of motion for $\Omega_i$ to be

$$I D_t \Omega_i = \epsilon_{ijk} M_j B_k + 2\Gamma \left( \omega_i - 2\Omega_i \right), \qquad (3)$$

where $I$ is the moment of inertia per particle, given by $I \sim r^2$, with $r$ being the typical radius of the ferrofluid particles. For the sake of simplicity, in our analysis, we assume the nanoparticles are spherical since the general case does not add much complexity. The governing equation for $M_i$ assumes that the magnetization is 'frozen' into the particle, and so the particles rotate as rigid objects,

$$D_t M_i = \epsilon_{ijk} \Omega_j M_k. \qquad (4)$$

A key assumption in ferrofluids is that the constituent particles are vanishingly small, and so we take the limit $I \to 0$. From Eq. (3) we see that this assumption implies that magnetic torque balances rotational friction

$$2\Gamma \left( \omega_i - 2\Omega_i \right) = -\epsilon_{ijk} M_j B_k. \tag{5}$$

We now solve Eq. (5) for $\Omega_i$ and substitute into (2) and (4), which gives

$$D_t v_i + \partial_i P = \mu \nabla^2 v_i + M_j \partial_j B_i + \frac{1}{2} \epsilon_{ijk} \partial_j \left( \epsilon_{klm} M_l B_m \right), \tag{6}$$

$$D_t M_i = \frac{1}{2} \epsilon_{ijk} \omega_j M_k - \frac{1}{4\Gamma} \epsilon_{ijk} M_j \left( \epsilon_{klm} M_l B_m \right). \tag{7}$$

Due to the smallness of the magnetic nanoparticles, thermal fluctuations can destroy the fluid magnetization for sufficiently low magnetic fields. A simplified model, appropriate for low magnetic field strengths, adds to Eq. (7) a dissipation term of the form $-(M_i - M_i^0)/\tau$, where $\tau$ is a characteristic relaxation time. The magnitude of the equilibrium magnetization $M_i^0$ is given in terms of the Langevin function

$$M_i^0 = \frac{\mu n B_i}{B} \left[ \coth \left( \frac{\mu B}{k_B T} \right) - \frac{k_B T}{\mu B} \right], \tag{8}$$

where $\mu$ is the magnetic moment of each particle and $n$ is the number of magnetic moments per unit of volume. In the absence of an externally applied field, or for high temperatures, the magnetic moments of each particle are 'randomized', and the magnetization $M_i^0$ vanishes [43, 46]. A more detailed analysis takes the particle orientations to be random variables governed by Eq. (3), and averages over all angles. Taking the limit $I \to 0$ and substituting $\langle \Omega_i \rangle_T$ into the equation for $M_i$ yields a similar dissipation term and modifies the rotational viscosity term [33, 37]. For this paper, either analysis will do, since the parity odd effects that we discuss are general and do not rely on specific microscopic details of the model.

The equations of motion above can also be derived starting from a Hamiltonian framework in combination with a dissipation function [45, 46]. Starting from a Hamiltonian of the form

$$H = \int d^3r \left[ \frac{1}{2\rho} g_i^c g_i^c + \frac{1}{2I\rho} \ell_i \ell_i - M_i B_i + U(\rho) \right] \tag{9}$$

and the non-vanishing Poisson brackets given by

$$\{ g_i^c(\vec{r}), \rho(\vec{r}') \} = \rho(\vec{r}) \, \partial_i \, \delta(\vec{r} - \vec{r}') , \tag{10}$$

$$\{ g_i^c(\vec{r}), g_j^c(\vec{r}') \} = \left[ g_j^c(\vec{r}) \, \partial_i - g_i^c(\vec{r}') \, \partial_j' \right] \delta(\vec{r} - \vec{r}') , \tag{11}$$

$$\{ g_i^c(\vec{r}), \ell_j(\vec{r}') \} = \ell_j(\vec{r}) \, \partial_i \, \delta(\vec{r} - \vec{r}') , \tag{12}$$

$$\{ g_i^c(\vec{r}), M_j(\vec{r}') \} = M_j(\vec{r}) \, \partial_i \, \delta(\vec{r} - \vec{r}') , \tag{13}$$

$$\{ \ell_i(\vec{r}), \ell_j(\vec{r}') \} = \epsilon_{ijk} \, \ell_k(\vec{r}) \, \delta(\vec{r} - \vec{r}') , \tag{14}$$

$$\{ \ell_i(\vec{r}), M_j(\vec{r}') \} = \epsilon_{ijk} \, M_k(\vec{r}) \, \delta(\vec{r} - \vec{r}') , \tag{15}$$

one can derive the equations of motion (2-4) using the following form of Hamilton's equations

$$\partial_t g_i^c = \{ g_i^c, H \} - \frac{\delta R}{\delta g_i^c}, \tag{16}$$

$$\partial_t \ell_i = \{ \ell_i, H \} - \frac{\delta R}{\delta \ell_i}, \tag{17}$$

$$\partial_t M_i = \{ M_i, H \} - \frac{\delta R}{\delta M_i}, \tag{18}$$

where the effects of viscosity and dissipation are included by the use of a dissipation function of the form

$$R = \int d^3r \left[ \frac{1}{2} \nu \left( \partial_i v_j + \partial_j v_i \right)^2 + \frac{1}{2} \Gamma \left( 2\Omega_i - \epsilon_{ijk} \partial_j v_k \right)^2 \right. \\ \left. + \frac{1}{2\tau} \left( M_i - M_i^0 \right)^2 \right]. \tag{19}$$

In deriving the equations of motion we use the relation between center of mass momentum density and velocity $g_i^c = \rho v_i$ (for brevity we simply use $v_i$ for the center of mass velocity), and the relation between angular momentum density and particle rotation $\ell_i = I\rho\Omega$. Equations of motion (16-18) with the choice of Hamiltonian (9) and dissipation function (19) lead to equations (2-4) with the aforementioned relaxation term $-(M_i - M_i^0)/\tau$ added to the magnetization dynamics (4).

Note that one may also add quadratic terms of the form

$$H_{MM} = \int d^3r \, \frac{1}{2} M_i M_i , \tag{20}$$

however with the brackets (11) - (15), this term only modifies the pressure definition in the in the incompressible limit. In this regime, the equation of state becomes singular and the pressure becomes fully determined by the flow. In other words, the terms coming from $H_{MM}$ do not contribute to the equations of motion.

In the standard ferrofluid analysis, the assumption of the vanishing moment of inertia per particle $I$ is imposed before setting $\tau \to 0$. In other words, the nanoparticle angular momenta relaxes much earlier than their magnetization. For a uniform magnetic field, the resulting equation is simply the Navier-Stokes equation

$$D_t v_i = \nu \nabla^2 v_i - \partial_i P , \tag{21}$$

together with the incompressibility condition $\partial_i v_i = 0$. If we write Eq. (21) in the form $D_t v_i = \partial_j T_{ij}$, and write the stress tensor in terms of a viscosity tensor $T_{ij} = -P\delta_{ij} + \eta_{ijkl}\partial_k v_l$, we have

$$\eta_{ijkl} = \nu \left( \delta_{ik}\delta_{jl} + \delta_{il}\delta_{jk} \right), \tag{22}$$

which is even under the exchange $ij \to kl$. *Thus, there is no parity breaking in this setup within standard ferrohydrodynamics.* We will reserve a more complete discussion on the matter to Sec. IV.

## III. PARITY BREAKING IN FERROFLUIDS

This section investigates under which conditions parity-breaking terms emerge in effective ferrofluid dynamics. The Hamiltonian and PB framework of ferrofluids provides a straightforward way to add modifications to the system. While many additional terms are in principle possible, not all terms are relevant or lead to interesting behavior.

### A. Absence of Parity Breaking in Ferrofluids with Vorticity-Angular Momentum Coupling

As an example, one may add the term seen in Markovich et al [24]

$$H_{\ell\omega} = \int d^3r \, \frac{1}{2}\ell_i\omega_i, \tag{23}$$

which couples fluid vorticity to intrinsic angular momentum density. This term is particularly relevant to the chiral active fluids of Soni et al [1], where the constituent particles are orders of magnitude larger than the conventional ferrofluids. While this does lead to parity-breaking terms in the stress tensor, they are washed out under the ferrofluid assumption of the vanishing moment of inertia. To show this, we now compute the governing equations resulting from the Hamiltonian $H + H_{\ell\omega}$, where $H$ is defined in Eq. (9). Using the brackets (10) - (15), along with the dissipation function (19), we obtain

$$D_t v_i = -\partial_i P + \nu\nabla^2 v_i + M_j\partial_i B_j + \frac{I}{2}\omega_j\partial_j\Omega_i$$
$$- \Gamma\epsilon_{ijk}\partial_j\left(\omega_k - 2\Omega_k\right), \tag{24}$$

$$ID_t\Omega_i = \epsilon_{ijk}M_jB_k + 2\Gamma\left(\omega_i - 2\Omega_i\right) + \frac{I}{2}\epsilon_{ijk}\omega_j\Omega_k$$
$$- \frac{I^2}{2}\epsilon_{jkm}\partial_j\Omega_i\partial_k\Omega_m, \tag{25}$$

$$D_t M_i = \epsilon_{ijk}\Omega_jM_k + \frac{1}{2}\epsilon_{ijk}\omega_jM_k - \frac{1}{\tau}(M_i - M_i^0)$$
$$- \frac{I}{2}\epsilon_{jkm}\partial_jM_i\partial_k\Omega_m. \tag{26}$$

Here we have already taken the incompressible limit, and for brevity have set $\rho = 1$. Upon taking the ferrofluid limit ($I \to 0$), Eq. (25) gives the same expression as in Eq. (5). *Therefore, the presence of the term (23) in the ferrofluid Hamiltonian does not modify the effective ferrohydrodynamic equations.* Physically this indicates that the angular momentum relaxes much faster than the characteristic time scales in which parity-odd effects can be observed.

### B. Parity Breaking in Ferrofluids from Vorticity-Magnetization Coupling

Magnetism in iron atoms occurs due to unpaired electrons in their electron configuration, which gives rise to a magnetic moment. Neglecting any contribution from the atomic nucleus, the atomic magnetic moment can be written as $g_J\mu_B\sqrt{j(j+1)}$, where $j$ is the total angular momentum quantum number, $g_J$ is the Landé g-factor, and $\mu_B$ is the Bohr magneton. From a quantum mechanics point of view, the total angular momentum of the ferrofluid particle can be split into molecular orbital and atomic angular momentum, which can be expressed in terms of the atom's magnetic moment. Therefore, under the replacement $\ell_i \to \ell_i + \gamma M_i$, where $\gamma \sim \frac{\hbar}{g_J\mu_B}$, the Hamiltonian term $H_{\ell\omega}$ becomes $H_{\ell\omega} + H_{M\omega}$, where $H_{M\omega}$ is defined in Eq. (1).

In the following, we will study the ferrofluid system described by the Hamiltonian

$$H = \int d^3r \left[\frac{1}{2\rho}g_i^c g_i^c + \frac{1}{2I\rho}\ell_i\ell_i + \frac{1}{2}\ell_i\omega_i + \frac{\gamma}{2}\omega_iM_i - M_iB_i\right]. \tag{27}$$

The above argument suggests that $\gamma \approx \frac{m_e}{e}$, however for the rest of our analysis we will leave the coupling constant $\gamma$ as a free parameter to be investigated experimentally. Even though the numerical value of $\gamma$ may be quite small, with a large enough external magnetic field the combination $\gamma M^0$, where $M^0$ is the equilibrium magnetization defined in Eq. (8), can in principle be arbitrarily large.

We now compute the governing equations resulting from the Hamiltonian (27), using the brackets (10) - (15), along with the dissipation function (19). We also take the incompressible limit and set $\rho = 1$ for brevity. We obtain

$$D_t v_i = -\partial_i P + \nu\nabla^2 v_i - \Gamma\epsilon_{ijk}\partial_j\left(\omega_k - 2\Omega_k\right)$$
$$+ M_j\partial_i B_j + \frac{1}{2}\omega_j\partial_j\left(I\Omega_i + \gamma M_i\right), \tag{28}$$

$$ID_t\Omega_i = 2\Gamma\left(\omega_i - 2\Omega_i\right) - \frac{I}{2}\epsilon_{jkm}\partial_j\Omega_i\partial_k\left(I\Omega_m + \gamma M_m\right)$$
$$+ \epsilon_{ijk}M_jB_k + \frac{1}{2}\epsilon_{ijk}\omega_j\left(I\Omega_k + \gamma M_k\right), \tag{29}$$

$$D_t M_i = \epsilon_{ijk}\Omega_jM_k + \frac{1}{2}\epsilon_{ijk}\omega_jM_k - \frac{1}{\tau}(M_i - M_i^0)$$
$$- \frac{1}{2}\epsilon_{jkm}\partial_jM_i\partial_k\left(I\Omega_m + \gamma M_m\right). \tag{30}$$

The terms involving $I\Omega_i + \gamma M_i$ stem from $H_{\ell\omega}$ and $H_{M\omega}$, while the second term on the right-hand side of (30) comes from $H_{\ell\omega}$ alone. The set of equations (28-30) model a wide class of magnetic fluids, of any particle size and any orientation of the magnetic field and magnetization.

Note that, even though we have associated the quantity $\gamma M_i$ to spin degrees of freedom, it algebra does not

satisfy the spin algebra, since $\{M_i(\vec{r}), M_j(\vec{r}')\} = 0$. If we wish to describe magnetic solids, or any motion of the magnetization relative to the particle orientation, we may add a nonvanishing bracket between $M_i$ and $M_j$, such that, $\{M_i(\vec{r}), M_j(\vec{r}')\} = \frac{1}{\gamma}\epsilon_{ijk} M_k(\vec{r})\,\delta(\vec{r} - \vec{r}')$. With the inclusion of this term, the equation of motion for $M_i$ becomes the Landau-Lifshitz-Gilbert equation, which would more precisely track the relaxation of the magnetization [48]. In this work, we assume the magnetization is 'frozen' into the ferrofluid particles. Physically this represents the fact that when attempting to align with the external magnetic field, it is more favorable for the particle to rotate than for the magnetization to rotate independently. This allows us keep $\{M_i(\vec{r}), M_j(\vec{r}')\} = 0$ throughout this analysis. This is justified when the phenomenological parameter $\gamma$ becomes large.

Upon taking the ferrofluid limit $(I \to 0)$, Eq. (29) gives the modified torque balance equation

$$2\Gamma\left(\omega_i - 2\Omega_i\right) + \epsilon_{ijk}M_j B_k + \frac{\gamma}{2}\epsilon_{ijk}\omega_j M_k = 0, \quad (31)$$

which shows that in the limit of vanishing particle size rotational friction is balanced not only by the magnetic torque but also by the torque coming from the misalignment of fluid vorticity and magnetization. Upon solving (31) for $\Omega_i$ and substituting into the rest of the equations of motion, we get the final form of the modified ferrofluid equations

$$D_t v_i = -\partial_i P + M_j \partial_i B_j + \nu \nabla^2 v_i + \frac{\gamma}{2}\omega_j \partial_j M_i$$
$$+ \epsilon_{ijk}\partial_j\left(\frac{1}{2}\epsilon_{klm}M_l B_m + \frac{\gamma}{4}\epsilon_{klm}\omega_l M_m\right), \quad (32)$$

$$D_t M_i = \epsilon_{ijk}\omega_j M_k - \frac{1}{4\Gamma}\epsilon_{ijk}M_j\left[\epsilon_{klm}M_l\left(B_m - \frac{\gamma}{2}\omega_m\right)\right]$$
$$- \frac{1}{\tau}\left(M_i - M_i^0\right) - \frac{\gamma}{2}\epsilon_{jkm}\partial_j M_i \partial_k M_m. \quad (33)$$

In the case of a uniform magnetic field applied in the positive $z$ direction, the small magnetic relaxation time $\tau$ sets $M_i = |\vec{M}^0|\delta_{iz}$, with $\vec{M}^0$ given by (8). Eq. (32) can then be written as

$$D_t v_i + \partial_i P = \nu \nabla^2 v_i + \frac{\gamma}{4}M^0 \partial_z \omega_i. \quad (34)$$

This last term breaks parity and is similar to the term obtained in [24] after taking the incompressible limit, albeit with $M_z$ playing the role of the intrinsic angular momentum $\ell_z = I\Omega_z$. We emphasize that this parity-breaking term is unique to three-dimensional fluids and has no analog in a 2D system. However, in the literature, the term 'odd viscosity' is used for certain parity-breaking terms in both 2D and 3D. In the following, we draw a comparison between the parity-breaking terms in 3D and 2D.

## IV. ANISOTROPIC VISCOSITY TENSOR IN 3D

The term $-\frac{1}{4}\gamma M^0$ is referred to in recent literature as 3D odd viscosity [24]. Before discussing the proposed Hele-Shaw experiment, it is convenient to clarify this terminology by briefly reviewing the form of the viscosity tensor for a 3D incompressible fluid with rotational invariance along one axis. In this section, this axis will be taken to be the $z$-axis for simplicity.

Note that, after the magnetization relaxes, the only dynamical variable in the system is the fluid velocity. In general, the incompressible hydrodynamic equation can be expressed as

$$\partial_i v_i = 0, \quad (35)$$
$$D_t v_j + \partial_j P = \partial_i\left(\eta_{ijkl}\partial_k v_l\right). \quad (36)$$

Taking the scalar product of Eq. (36) with $v_i$ and using the incompressibility condition (35), we find the energy equation:

$$\partial_t\left(\tfrac{1}{2}v_j^2\right) + \partial_i\left[\left(\tfrac{1}{2}v_j^2 + P\right)v_i - \eta_{ijkl}v_j\partial_k v_l\right]$$
$$= -\eta_{ijkl}\partial_i v_j \partial_k v_l. \quad (37)$$

Since the left-hand side is a quadratic form, energy is dissipated when $\eta_{ijkl} = \eta_{klij} > 0$. However if $\eta_{ijkl} = -\eta_{klij}$, the system is non-dissipative, and the sign of $\eta_{ijkl}$ is not determined.

For an isotropic incompressible system, there are only two viscosity coefficients, namely

$$\eta_{ijkl} = \nu\left(\delta_{ik}\delta_{jl} + \delta_{il}\delta_{jk}\right) + \Gamma\epsilon_{ijm}\epsilon_{klm}. \quad (38)$$

Note that, both $\nu$ and $\Gamma$ are not invariant under time reversal, but they do not break parity symmetry. All components of $\eta_{ijkl}$ break time reversal, however, only some of them lead to dissipation. In general, terms that are odd under time reversal and do not spoil energy conservation are also odd under parity symmetry, as long as they are invariant under the combined $PT$-symmetry.

So, instead of focusing on a general viscosity tensor, we will only focus on $PT$-symmetric terms. That is, the coefficients such that $\eta_{ijkl} = -\eta_{klij}$. In a fully isotropic system, all of these coefficients vanish, however, if we impose rotational invariance only on the $xy$ plane we find that the viscosity tensor admits non-zero coefficients. Furthermore, since we have assumed that the angular momentum has already relaxed, one should expect the stress tensor to be symmetric, which imposes that $\eta_{ijkl} = \eta_{jikl}$. There are only two viscosity coefficients that satisfy all these conditions, i.e., $\eta_{ijkl} = -\eta_{klij} = \eta_{jikl}$, together with rotational invariance in the $xy$-plane. They are

$$\eta_{ijkl}^{PT}\partial_k v_l = \nu_o^\perp\left[\epsilon_{ikz}\left(\delta_{jm} - \delta_{jz}\delta_{mz}\right)\partial_k v_m\right.$$
$$+ \epsilon_{jkz}\left(\delta_{im} - \delta_{iz}\delta_{mz}\right)\partial_m v_k\right]$$
$$+ \nu_o^\parallel\left(\epsilon_{ikz}\delta_{jz} + \epsilon_{jkz}\delta_{iz}\right)\left(\partial_k v_z + \partial_z v_k\right). \quad (39)$$

As shown in Ref. [21], $\nu_o^\perp$ and $\nu_o^\parallel$ are also the two viscosity coefficients that remain non-zero in the collisionless limit

of a relativistic plasma subjected to an external magnetic field.

The term multiplying $\nu_o^\perp$ has the same form as the 2D odd viscosity stress tensor. The coefficients $\nu_o^\perp$ and $\nu_o^\parallel$ are in principle unrelated, however, for relativistic collisionless plasma in the magnetic field, $\nu_o^\parallel = -\frac{1}{2}\nu_o^\perp$. This relation is also observed in Ref. [24], which indicates that it is a sufficient condition for Hamiltonian systems. In addition, this relation seems to simplify substantially the expression for the lift force on a circular object inside an incompressible flow with parity-odd viscosity coefficients, as pointed out in Ref. [32].

Under assumption $\nu_o^\parallel = -\frac{1}{2}\nu_o^\perp$, the term $\partial_z\omega_i$ in the momentum equation can be associated to the in-plane odd viscosity. In principle, for $\nu_o^\perp \neq 2\nu_o^\parallel$, we have

$$\partial_i\left(\eta_{ijkl}^{PT}\partial_k v_l\right) = \left(2\nu_o^\parallel + \nu_o^\perp\right)\left(\partial_j\partial_z v_z + \delta_{jz}\partial_z\omega_z\right)$$
$$- \nu_o^\perp\partial_j\omega_z - \nu_o^\parallel\partial_z\omega_j\,, \quad (40)$$

for incompressible fluids. The term $-\nu_o^\perp\partial_j\omega_z$ modifies the fluid pressure, whereas $-\nu_o^\parallel\partial_z\omega_j$ correspond to the extra term in Eq. (34), if we set $\nu_o^\parallel = -\frac{1}{4}\gamma M^0$. Therefore, in this case, referring to parity-odd viscosity coefficients by 3D odd viscosity is only accurate upon the identification $\nu_o^\perp = -2\nu_o^\parallel$.

## V. FERROFLUIDS IN HELE-SHAW CELL: EXPERIMENTAL PROPOSAL

A Hele-Shaw (HS) cell is a simplified experimental setup that could potentially test the presence of vorticity-magnetization coupling leading to parity-breaking effects in ferrofluids. A HS cell consists of two parallel plates separated by an infinitesimally small gap $h$, with the hydrodynamic variables assumed to vary in the $xy$ plane at much larger length scales than the distance between the two plates. Confining ferrofluids to a HS cell has much experimental and theoretical background [49–52]. A remarkable example is that of the labyrinthine instability [40]. For our purposes, we use the HS cell as a way to measure this new coupling constant $\gamma$.

Following the analysis seen in [2], we take the coordinates to scale as

$$t \sim \frac{1}{\epsilon}\,, \qquad x \sim \frac{1}{\epsilon}\,, \qquad y \sim \frac{1}{\epsilon}\bar{y}\,, \qquad z \sim \epsilon^0\,, \quad (41)$$

where $\epsilon = h/L$, with $L$ being some characteristic length in the $xy$ plane. The hydrodynamic variables scale as

$$P \sim \frac{1}{\epsilon}\,, \quad v_x \sim \epsilon^0\,, \quad v_y \sim \epsilon^0\,, \quad v_z \sim \epsilon\,. \quad (42)$$

Using the relaxed form of our system, Eq. (34), the governing equations become

$$0 = \partial_x v_x + \partial_y v_y + \partial_z v_z\,, \quad (43)$$

$$\partial_x P = \nu\partial_z^2 v_x - \frac{\gamma}{4}M^0\partial_z^2 v_y\,, \quad (44)$$

$$\partial_y P = \nu\partial_z^2 v_y + \frac{\gamma}{4}M^0\partial_z^2 v_x\,, \quad (45)$$

$$\partial_z P = 0\,, \quad (46)$$

The solutions for velocity that satisfy the no-slip and no-penetration boundary conditions on the plates are

$$v_x = \frac{6z}{h^2}(h - z)V_x(x, y)\,, \quad (47)$$

$$v_y = \frac{6z}{h^2}(h - z)V_y(x, y)\,, \quad (48)$$

$$v_z = 0\,, \quad (49)$$

and pressure is now independent of $z$. We then substitute these solutions into (43) - (45), which gives

$$0 = \partial_x V_x + \partial_y V_y\,, \quad (50)$$

$$-\frac{h^2}{12}\partial_x P = \nu V_x - \frac{\gamma}{4}M^0 V_y\,, \quad (51)$$

$$-\frac{h^2}{12}\partial_y P = \nu V_y + \frac{\gamma}{4}M^0 V_x\,, \quad (52)$$

The above set of equations is Darcy's law, the standard governing equation of flow within an HS cell, with an extra term coming from $M^0$. This equation is of the same form as that seen in Ref. 2, with $\gamma M^0/4$ playing the role of the parity odd parameter seen there.

The experimental setups discussed in Ref. 2 provide methods to measure this coupling. Here we will only describe a simplified setup that highlights a key feature of parity odd flows. Consider HS cell with a central injection point, where a fluid (or air) is injected with a constant rate $q$ (see Fig 1). Details of the inside fluid are not important for this discussion, since the following effect is independent of any intricate interface features or instabilities.

Equations (50-52) indicate that the ferrofluid flow is irrotational and the pressure is a harmonic function, that is,

$$\partial_x V_y - \partial_y V_x = 0\,, \quad (53)$$
$$\nabla^2 P = 0\,. \quad (54)$$

It is not hard to see that $V(\zeta) \equiv V_x(x, y) - iV_y(x, y)$, with $\zeta = x + iy$, is an analytic complex function, since the Cauchy-Riemann conditions become the incompressibility and the irrotational equations (50,53). Assuming that the ferrofluid domain $\Omega(t)$ starts at the fluid interface and extends out to infinity, the analytic velocity function $V(\zeta)$ can be expanded in the Laurent series as

$$V(\zeta) = \sum_{n=1}^{\infty}\frac{c_n}{\zeta^n}\,, \quad (55)$$

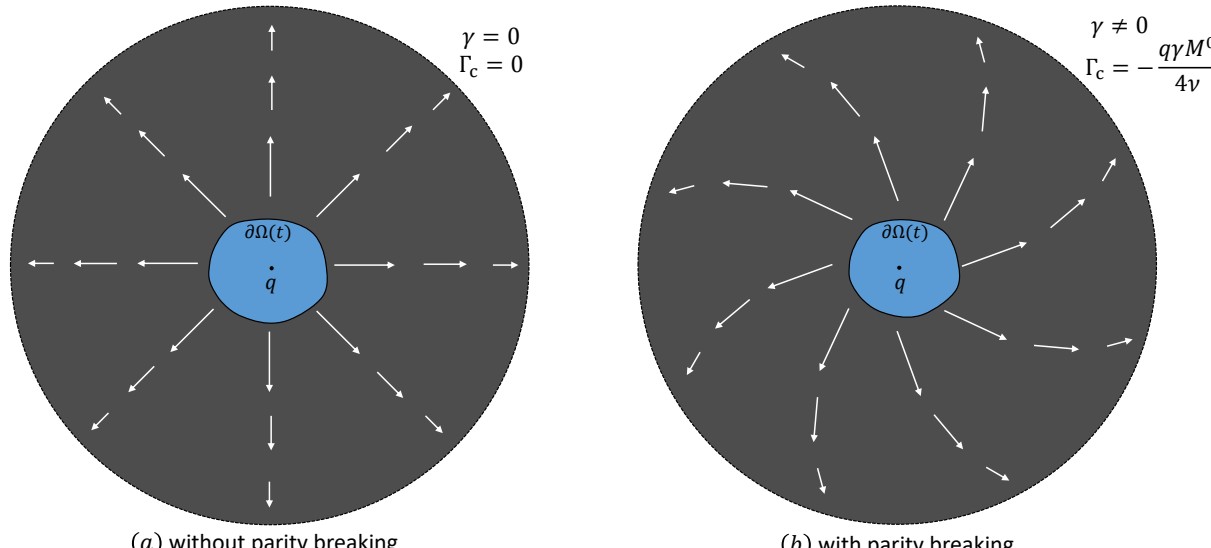

Figure 1. Schematic picture of injection of a test fluid (blue) into a ferrofluid (black), with a distinct interface $\partial\Omega$ formed between them. (a) In the absence of the parity-breaking coupling $\gamma$, the ferrofluid flows radially outward and has no far field circulation ($\Gamma = 0$). (b) When the parity-breaking coupling $\gamma$ is nonzero the ferrofluid acquires a spiral behavior everywhere in the domain. In the far field, the circulation can be computed even without knowing the details of the interface.

where the coefficients $c_n$ are determined by the boundary conditions at the fluid interface. From the residue theorem, one can determine the coefficient $c_1$ by integrating $V(\zeta)$ over the fluid interface $\partial\Omega$, i.e,

$$2\pi i c_1 = \oint_{\partial\Omega} V(\zeta)d\zeta = \Gamma_C - i\frac{d\mathcal{A}}{dt}, \qquad (56)$$

where $\Gamma_C$ is the fluid circulation and $d\mathcal{A}/dt = -q$ is the rate of change of the area occupied by the ferrofluid.

From the expression (55) it is possible to determine the ferrofluid pressure, which becomes

$$P = -\frac{3}{h^2}\,\text{Re}\left[(4\nu - i\gamma M^0)\left(\frac{q - i\Gamma_C}{2\pi}\ln\zeta - \sum_{n=1}^{\infty}\frac{c_{n+1}}{n\zeta^n}\right)\right]. \qquad (57)$$

Since the pressure must be a single-valued function, this imposes that $(4\mu - i\gamma M^0)(q - i\Gamma_C)$ must be a real coefficient. This shows that the ferrofluid circulation must be proportional to

$$\Gamma_C = -\frac{q\gamma M^0}{4\nu}. \qquad (58)$$

The existence of non-zero circulation in the outer region (far field flow of ferrofluids) can be used to directly measure the coupling constant $\gamma$ and serves as a straightforward way to test for the presence or absence of the vorticity-magnetization coupling in ferrofluids. We emphasize that the far-field circulation, represented by the variable $\Gamma$, is independent of the details of the two-fluid interface. The physics at the boundary between two fluids can vary greatly depending on the viscosity of the fluid being injected. For example, when a less viscous

fluid such as an air bubble is injected into a ferrofluid, the Saffman-Taylor fingering instabilities can occur. These instabilities are not present when a more viscous fluid is injected. The advantage of this proposal is that the circulation in the far field, away from the interface, is not affected by the intricate and complex physics that occurs near the interface.

## VI. CONCLUSION

In this work, we introduced a new coupling in ferrofluids between fluid vorticity and magnetization. This modification is introduced in the context of a Hamiltonian and PB framework, and after following the standard ferrofluid assumptions we arrive at a modified set of governing equations containing extra parity-odd terms. When confined to a HS cell, these extra terms manifest as off-diagonal components of Darcy's Law and provide a robust experimental setting in which to measure this new coupling.

An important conclusion from our analysis is that simply including $H_{\ell\omega}$ is not enough to capture the parity-odd effects in ferrofluids due to the small size of the particles. Additionally, the coarse-graining procedure outlined in Ref. [24] can be used to determine the specifics of $\gamma$ from a microscopic perspective. Future research in this direction will depend heavily on experimental confirmation of the existence of parity-breaking terms in ferrofluids.

## VII. ACKNOWLEDGMENTS

We thank J.C. Burton for useful discussions. This work is supported by NSF CAREER Grant No. DMR-1944967 (SG). DR is supported by the 21st Century Foundation Startup award from CCNY. GMM was supported by the National Science Foundation under Grant OMA-1936351.

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
