# Peer review of "Parity Breaking in Ferrofluids with Vorticity-Magnetization Coupling"

_SciPost Physics_

## Round 1 · Referee Report · Anonymous (Referee 1) · 2023-11-13

Strengths

  1. The manuscript contains new results. These are the system equations for a ferrofluid with an additional vorticity-magnetization coupling. The Hamiltonian structure of these equations is also presented.

  2. The authors propose an experiment which would demonstrate the presence (or absence) of the introduced new term by measuring the circulation of the fluid in a particular Hele-Shaw cell setup.

  3. The manuscript is nicely written

  4. It contains references to relevant literature

Weaknesses

While an additional term proposed by the authors is allowed by symmetry in parity broken fluids, it is not clear what one should expect for the value of the introduced coupling $\gamma$. The authors say that it would depend on the "microscopic properties of the fluid nanoparticles". However, it is not clear whether it can be significant for small size particles. They also say "a heuristic analysis of the structure of the ferrofluid nanoparticles and the atoms within, shows $\gamma$ to be inversely proportional to the molecule Lande g-factor." I think it would be useful for readers if the authors expanded that comment and give that heuristic argument in slightly more detail.

Report

The manuscript introduces an additional symmetry-allowed term into the ferrofluid hydrodynamic equations. They study the effect of the term on the fluid motion and describe a possible test for the presence of the term in the Hele-Shaw cell experiment.

I think that the manuscript is interesting and it contains new results. In my opinion, it should be published in SciPost Physics with a few minor changes described below. (the changes are minor and do not affect any of the conclusions of the authors)

Requested changes

  1. Expand the comment on ``heuristic argument'' to make it more clear (see above).

  2. Reference in abstract: it is better to have a numbered citation their [32] instead of the full citation which is hard to find in the list of references.

  3. The second paragraph in the first column. The comparison to "incompressible fluids" is too abrupt and unnecessary as the active chiral fluids exhibit additional transport coefficients even in comparison with compressible fluids.

  4. The second paragraph in the second column on the first page. "A ferrofluid is a type of 3D active matter fluid...". This is not strictly correct. Only driven ferrofluids should be considered as active. The main example of the paper, the fluid with constant magnetization, for example, is not active.

  5. After equation (8). I believe that there should be an equilibrium magnetization $M_i^0$ in the sentence "... and the magnetization $M_i$ vanishes..."

  6. After eq. (21) "to" is missing in "does lead parity breaking terms"

  7. Just above (41). The sentence "... and average over the plate separation..." is not clear. I think the authors meant "... and average over the thickness of the fluid between the plates..." or something similar. Please, clarify.

  8. Right before the end of the first column on the page 6. "the two fluid interface." should be "the two-fluid interface."

  • validity: high
  • significance: high
  • originality: high
  • clarity: top
  • formatting: perfect
  • grammar: excellent

Author:  Dylan Reynolds  on 2024-07-18  [id 4631]

(in reply to Report 1 on 2023-11-13)

Response to Referee 1

Strengths:

  1. The manuscript contains new results. These are the system equations for a ferrofluid with an additional vorticity-magnetization coupling. The Hamiltonian structure of these equations is also presented.
  2. The authors propose an experiment that would demonstrate the presence (or absence) of the introduced new term by measuring the circulation of the fluid in a particular Hele-Shaw cell setup.
  3. The manuscript is nicely written
  4. It contains references to relevant literature

Weaknesses:

While an additional term proposed by the authors is allowed by symmetry in parity broken fluids, it is not clear what one should expect for the value of the introduced coupling γ. The authors say that it would depend on the "microscopic properties of the fluid nanoparticles”. However, it is not clear whether it can be significant for small size particles. They also say "a heuristic analysis of the structure of the ferrofluid nanoparticles and the atoms within, shows γ to be inversely proportional to the molecule Lande g-factor.” I think it would be useful for readers if the authors expanded that comment and give that heuristic argument in slightly more detail.

Report:

The manuscript introduces an additional symmetry-allowed term into the ferrofluid hydrodynamic equations. They study the effect of the term on fluid motion and describe a possible test for the presence of the term in the Hele-Shaw cell experiment. I think that the manuscript is interesting and it contains new results. In my opinion, it should be published in SciPost Physics with a few minor changes described below. (the changes are minor and do not affect any of the conclusions of the authors)

Requested changes:

  1. Expand the comment on “heuristic argument” to make it more clear (see above).

DR, GMM, and SG: We have expanded our heuristic argument by providing a more in-depth analysis of the microscopics.

  1. Reference in abstract: it is better to have a numbered citation their [2] instead of the full citation which is hard to find in the list of references.

DR, GMM, and SG: Agree and fixed.

  1. The second paragraph in the first column. The comparison to "incompressible fluids” is too abrupt and unnecessary as the active chiral fluids exhibit additional transport coefficients even in comparison with compressible fluids.

DR, GMM, and SG: Even though ferrofluids are typically considered to be incompressible, we believe that referee has a valid point. We changed the Poisson brackets to account for the fluid density and the incompressible limit is then taken at the level of the equations of motion.

  1. The second paragraph in the second column on the first page. "A ferrofluid is a type of 3D active matter fluid...”. This is not strictly correct. Only driven ferrofluids should be considered as active. The main example of the paper, the fluid with constant magnetization, for example, is not active.

DR, GMM, and SG: We have restructured this paragraph to avoid this statement and to more clearly state what we investigate.

  1. After equation (8). I believe that there should be an equilibrium magnetization M 0 i in the sentence "... and the magnetization Mi vanishes...”

DR, GMM, and SG: Fixed.

  1. After eq. (21) "to” is missing in "does lead parity breaking terms”

DR, GMM, and SG: Fixed.

  1. Just above (41). The sentence "... and average over the plate separation...” is not clear. I think the authors meant "... and average over the thickness of the fluid between the plates...” or something similar. Please, clarify.

DR, GMM, and SG: Upon redoing the calculation, we find that no averaging is necessary. We simply insert Eq. (47-49) into the system (43-45), which gives the desired result.

  1. Right before the end of the first column on the page 6. "the two fluid interface.” should be "the two-fluid interface.”

DR, GMM, and SG: Fixed. We thank the referee for their helpful feedback in polishing the manuscript.

---

## Round 1 · Referee Report · Anonymous (Referee 2) · 2023-11-19

Strengths

1- The paper is well written 2- Odd viscous phenomena are explored theoretically in a physical system where it has not been explored previously 3- The experimental proposal is concrete and simple

Weaknesses

1- As the authors acknowledge, the overlap with reference [22] is very strong. 2-The fluid mechanical results that lead to the experimental proposal that are presented also hold for general odd viscous systems and these have to a large extent been presented earlier by the same authors in [32]. Mostly what this paper does is implicitly restrict to a certain odd viscosity that happens to coincide with ferrofluids, but also with fluids with an intrinsic angular momentum-vorticity coupling. 3- Apart from the overlap with [22], there is also strong overlap with plasmas that also produce odd viscosity when subjected to a background magnetic field. 4- One of the main strengths is that a new physical system is being explored through the lens of odd viscosity, but little details about these systems are provided.

Report

The authors study a vorticity-magnetization coupling called \gamma which is assumed to be part of the theoretical description of ferrofluids and which they show, with a Poisson bracket formalism, gives rise to a parity breaking term which at least from the title and introduction I assume is considered to be an odd viscous term. They then study the implications of this term for the Hele-Shaw cell.

Throughout the work, the resemblance with reference [22] is stressed, where the main difference is that for [22] there is a coupling between vorticity and intrinsic angular momentum. Below (31) it is stated that the final parity-breaking term that comes from the \gamma term is identical to the momentum balance law with odd viscosity obtained in [22] after taking the incompressible limit, I have some comments about this
1) Despite that the incompressible limit simplifies it to this single term, it would still to understand the extent to which this term can be seen as originating from odd viscosity. This is also important considering that the title of the work involves odd viscosity but the word odd viscosity is then not used at all after the introduction.
2) A previous work by the same authors [32] considered the Hele-Shaw problem for general 3D odd viscosities, so this stresses that more elaboration on (31) would be helpful to understand what is new and what isn’t. I do see that below (43) there is a statement about how there is a relation to the odd parameter of [32] but if (31) would be analyzed more thoroughly from the point of view of odd viscosity this would not come as a surprise since [32] works with general odd viscosities.
3) Is there anything known in the literature about the degree to which ferrofluids are incompressible? It would be nice if there were a discussion of this since it clearly has implications.
Furthermore, as mentioned in 2), it seems that an important part of what is novel about the work is that the authors utilize somewhat known derivations in the context of a different system, namely ferrofluids. Therefore, it would be helpful if more details would be given about ferrofluids. I see some statements below equation (24) about a “heuristic analysis of the structure of the ferrofluid nanoparticles”. This is a great opportunity to provide more details about the nature of ferrofluids, maybe with some numbers that give a sense of what an experimental setup of ferrofluids that could display the phenomena described in this work would look like. The authors also mention that the Hele-Shaw cell is considered by experimentalists in the context of ferrofluids and provide references. Is there anything about what these experimentalists have reported that is worth mentioning because it can perhaps be related to the \gamma term and odd viscosity?
Another key result presented in this work is the relation between the far field rotation and the \gamma term. Also here it would be good to have an understanding of how general the relation between 3d odd viscosity and far field circulation is. Particularly, in [32] a general anisotropic 3d system with odd viscosity is considered and a relation between circulation and the change of a bubble area is presented. Is there anything different here except that a specific odd viscosity is considered?
Lastly, the similarity with [22] is stressed in the sense that the equations would be equivalent if magnetization were replaced with intrinsic angular momentum. However, this also seems very similar to plasmas subject to an ordinary background magnetic field. This connection is made in [22] where it is stated that intrinsic angular momentum which induces odd viscosity could be replaced by a background magnetic field so that it coincides with what was found by Landau-Lifshitz. One could ask whether there is a difference between plasmas subject to a background magnetic field and fluids with magnetization that is turned on by a background magnetic field.
I think that the message that intrinsic angular momentum can be replaced by magnetization and that this allows for a result that follows from a computation identical to [22] is an interesting one but that this alone is not novel enough to warrant publication. With additional fluid mechanical results related to experimental implications the situation could be different, however it seems that the fluid mechanical results currently presented are merely restricted versions of general fluid mechanical results already presented in [32]. I also believe that currently this work does not make a great case for why experimentalists should look for odd viscosity specifically in ferrofluids as opposed to all the other fluids that can potentially display odd viscosity.

Requested changes

1- Make it transparent that the resulting odd term is a term that could be written as an odd viscosity in the stress tensor, and therefore that the fluid mechanical results that were obtained earlier for general 3d odd viscosities can be obtained by restricting to this odd viscosity. 2- Then highlight anything new with respect to these previously obtained fluid mechanical results. 3- Provide more details about what the state of fluid mechanical experiments for ferrofluids is and if there is anything that suggests that odd viscosity could be observed. 4-Comment on magnetized plasmas.

  • validity: top
  • significance: good
  • originality: low
  • clarity: high
  • formatting: perfect
  • grammar: perfect

Author:  Dylan Reynolds  on 2024-07-18  [id 4635]

(in reply to Report 2 on 2023-11-19)

Response to Referee 2

Strengths:

1- The paper is well written 2- Odd viscous phenomena are explored theoretically in a physical system where it has not been explored previously 3- The experimental proposal is concrete and simple

Weaknesses:

1- As the authors acknowledge, the overlap with reference [24] is very strong.

DR, GMM, and SG: Ferrofluids possess many features that break parity at the microscopic scale, but it is puzzling why, at the macroscopic scale, most observables seem to show no signature of parity breaking. Motivated by this puzzle, the main question we are addressing in this manuscript is: what are the conditions for a ferrofluid to exhibit the parity-odd phenomena studied in Ref. [2]? In Ref. [24], the authors proposed a mechanism that generates the 3D odd viscosity effects in a fluid upon the relaxation of its intrinsic angular momentum. However, we showed in this manuscript that the coupling introduced in Ref. [24] does not lead to parity breaking (odd viscosity) in ferrofluids due to the small particle size. This mechanism works for large particles, like the magnetic colloids where the magnetization relaxes much faster than the intrinsic angular momentum of the constituents, which is not the case in ferrofluids. Motivated by our previous paper [2], we investigated which equivalent mechanism could bring 3D odd viscosity effects to ferrofluids. Additionally, the Hele-Shaw setup provides a robust experimental test that allows our theory to be verified. Indicating whether such a mechanism is present or not in ferrofluids.

2-The fluid mechanical results that lead to the experimental proposal that are presented also hold for general odd viscous systems and these have to a large extent been presented earlier by the same authors in [2]. Mostly what this paper does is implicitly restrict to a certain odd viscosity that happens to coincide with ferrofluids, but also with fluids with an intrinsic angular momentum-vorticity coupling.

DR, GMM, and SG: We would like to emphasize that the intrinsic angular momentum-vorticity coupling is not responsi- ble for parity-odd effects in ferrofluids. This is the main point of the paper, and as such we have added a new subsection outlining the distinction. The coupling term introduce in Ref. [24] does not capture the ferrofluid physics with parity breaking terms. To observe any of the effects predicted in [2], we need the addition of this rather unusual vorticity magnetization in the ferrofluid Hamiltonian.

3- Apart from the overlap with [24], there is also strong overlap with plasmas that also produce odd viscosity when subjected to a background magnetic field.

DR, GMM, and SG: Indeed, to the best of our knowledge, the 3D odd viscosity was first predicted in plasma systems subjected to magnetic fields. It was even discussed in Landau and Lifshitz, Physical Kinetics, vol. 10. We have added brief comments throughout the paper that address the relation.

4- One of the main strengths is that a new physical system is being explored through the lens of odd viscosity, but little details about these systems are provided.

DR, GMM, and SG: Even though the 3D odd viscosity is well-known and shows up in different areas, the scope of the paper is to focus on whether this term can be present in ferrofluids since they are natural candidates to be studied in a Hele-Shaw setup. So far, despite of their overall magnetization, parity-breaking terms were not considered in the hydrodynamics of ferrofluids.

Report:

The authors study a vorticity-magnetization coupling called γ which is assumed to be part of the theoretical description of ferrofluids and which they show, with a Poisson bracket formalism, gives rise to a parity-breaking term which at least from the title and introduction I assume is considered to be an odd viscous term. They then study the implications of this term for the Hele-Shaw cell. The resemblance with reference [24] is stressed throughout the work, where the main difference is that for [24] there is a coupling between vorticity and intrinsic angular momentum. Below (31) it is stated that the final parity-breaking term that comes from the γ term is identical to the momentum balance law with odd viscosity obtained in [24] after taking the incompressible limit, I have some comments about this

1) Despite the incompressible limit simplifies it to this single term, it would still be to understand the extent to which this term can be seen as originating from odd viscosity. This is also important considering that the title of the work involves odd viscosity but the word odd viscosity is then not used at all after the introduction. 2) A previous work by the same authors [2] considered the Hele-Shaw problem for general 3D odd viscosities, so this stresses that more elaboration on (31) would be helpful to understand what is new and what isn’t. I do see that below (43) there is a statement about how there is a relation to the odd parameter of [22] but if (31) were analyzed more thoroughly from the point of view of odd viscosity this would not come as a surprise since [22] works with general odd viscosities.

DR, GMM, and SG: Unfortunately, the term odd viscosity is used broadly in the literature. For purely 2D dimensional systems, the odd viscosity term corresponds to

$$ \tau_{ij}^\perp=\nu_o^\perp\left[\epsilon_{ikz}\left(\delta_{jm}-\delta_{jz}\delta_{mz}\right)\partial_kv_m+\epsilon_{jkz}\left(\delta_{im}-\delta_{iz}\delta_{mz}\right)\partial_mv_k\right]. $$
In 3D, there is an additional non-dissipative viscosity coefficient, which comes from the term
$$ \tau_{ij}^\|=\nu_o^\|\left(\epsilon_{ikz}\delta_{jz}+\epsilon_{jkz}\delta_{iz}\right)\left(\partial_kv_z+\partial_zv_k\right). $$
The coefficients $\nu_o^\perp$ and $\nu_o^\|$ are in principle unrelated, but in a Hamiltonian system, they must satisfy
$$ \nu_o^\perp=-2\nu_o^\|\,. $$
This was previously pointed out in Refs.~[24, 25, 32]. Only under this assumption, the term $\partial_z\omega_i$ in the momentum equation can be associated with the in-plane odd viscosity. In principle, for $\nu_o^\perp\neq 2\nu_o^\|$, we have
$$ \partial_i\left(\tau_{ij}^\perp+\tau_{ij}^\|\right)=\left(2\nu_o^\|+\nu_o^\perp\right)\left(\partial_j\partial_z v_z+\delta_{jz}\partial_z\omega_z\right)-\nu_o^\perp\partial_j\omega_z-\nu_o^\|\partial_z\omega_j\,, $$
for incompressible fluids. This is the main reason we avoided the term odd viscosity in most of the main text. We included this discussion in the new version of the manuscript.

3) Is there anything known in the literature about the degree to which ferrofluids are incompressible? It would be nice if there were a discussion of this since it clearly has implications. Furthermore, as mentioned in 2), it seems that an important part of what is novel about the work is that the authors utilize somewhat known derivations in the context of a different system, namely ferrofluids. Therefore, it would be helpful if more details would be given about ferrofluids. I see some statements below equation (24) about a “heuristic analysis of the structure of the ferrofluid nanoparticles”. This is a great opportunity to provide more details about the nature of ferrofluids, maybe with some numbers that give a sense of what an experimental setup of ferrofluids that could display the phenomena described in this work would look like. The authors also mention that the Hele-Shaw cell is considered by experimentalists in the context of ferrofluids and provide references. Is there anything about what these experimentalists have reported that is worth mentioning because it can perhaps be related to the γ term and odd viscosity?

DR, GMM, and SG: The incompressible approximation of ferrofluids is well-established experimentally and matches the theoretical framework of incompressible ferrohydrodynamics (see Ref. [33]). There is a comprehensive book by Rosensweig (Ref. [33]) on ferrohydrodynamics, where details of ferrofluids are discussed in depth. We have reviewed the pertinent theoretical aspects of ferrofluids in Section 2. Another key result presented in this work is the relation between the far field rotation and the γ term. Also here it would be good to have an understanding of how general the relation between 3d odd viscosity and far field circulation is. Particularly, in [2] a general anisotropic 3d system with odd viscosity is considered and a relation between circulation and the change of a bubble area is presented. Is there anything different here except that a specific odd viscosity is considered?

DR, GMM, and SG: The far-field circulation was worked out in detail in Ref. [2]. The only assumption there was the existence of parity-odd terms in the viscosity tensor. In this manuscript, we studied the conditions for this effect to be observed in ferrofluids since they are natural candidates to be placed in a Hele-Shaw cell and parity-odd effects have been overlooked in these systems. Once again, we would like to point out that the term odd viscosity is used too broadly in the literature. Referring to parity-odd viscosity coefficients by 3D odd viscosity is only accurate upon the identification $\nu_o^\perp=-2\nu_o^\|$. To further clarify this aspect, we have explicitly mentioned that $\nu_o^\perp=-2\nu_o^\|=-\tfrac{1}{4}\gamma M^0$ in the revised version of the manuscript.

Lastly, the similarity with [24] is stressed in the sense that the equations would be equivalent if magnetization were replaced with intrinsic angular momentum. However, this also seems very similar to plasmas subject to an ordinary background magnetic field. This connection is made in [24] where it is stated that intrinsic angular momentum which induces odd viscosity could be replaced by a background magnetic field so that it coincides with what was found by Landau-Lifshitz. One could ask whether there is a difference between plasmas subject to a background magnetic field and fluids with magnetization that is turned on by a background magnetic field I think that the message that intrinsic angular momentum can be replaced by magnetization and that this allows for a result that follows from a computation identical to [24] is an interesting one but that this alone is not novel enough to warrant publication. With additional fluid mechanical results related to experimental implications, the situation could be different, however it seems that the fluid mechanical results currently presented are merely restricted versions of general fluid mechanical results already presented in [2]. I also believe that currently, this work does not make a great case for why experimentalists should look for odd viscosity specifically in ferrofluids as opposed to all the other fluids that can potentially display odd viscosity.

DR, GMM, and SG: We would like to clarify some points raised by the referee: 1) The intrinsic angular momentum and magnetization have completely different microscopic origins. Focusing solely on the mathematical structure of these terms is useful for analogies, but many times overlooks their distinct underlying physical mechanisms. In particular, magnetization here is a dynamic field, unlike the background magnetic fields in plasmas. The core point we are addressing in this manuscript is what are the conditions for a ferrofluid to exhibit the phenomena studied in Ref. [2]. As we showed in this manuscript, the coupling term introduced in Ref. [24] does not generate any parity-odd effect on ferrofluids, since the small size of the ferrofluid particles washes them out. The terms ℓiωi and Miωi do share the same mathematical structure, but they correspond to distinct microscopic systems and different ferrohydrodynamic equations. Moreover, if observed, the presence of parity-breaking effects in ferrofluids begs the question of the microscopic origin of such coupling.

2) Ref. [2] introduces the investigation of parity-breaking terms in 3D fluids within the Hele-Shaw framework. This work inspired us to explore this path, as ferrofluids are the most commonly studied fluids in the Hele-Shaw setup. Our present study investigates whether and how parity breaking manifests in ferrofluids, a topic not previously addressed in ferrofluid literature. If evidence of parity breaking is found in ferrofluids, they could provide a simple ’tabletop’ experiment within a Hele-Shaw cell, avoiding the need for complex or active fluids.

Requested changes: 1- Make it transparent that the resulting odd term is a term that could be written as an odd viscosity in the stress tensor, and therefore that the fluid mechanical results that were obtained earlier for general 3d odd viscosities can be obtained by restricting to this odd viscosity.

DR, GMM, and SG: We fixed this and explicitly presented the expression for the odd viscosity in terms of the ferrofluid magnetization.

2- Then highlight anything new with respect to these previously obtained fluid mechanical results. 3- Provide more details about what the state of fluid mechanical experiments for ferrofluids is and if there is anything that suggests that odd viscosity could be observed.

DR, GMM, and SG: Our aim of this paper is to provide a first principles/theoretical framework for our new coupling. We leave details of ferrofluid experiments to future work.

4-Comment on magnetized plasmas.

DR, GMM, and SG: To our knowledge, magnetized plasmas are not studied within a Hele-Shaw setup and hence we would like to keep the discussion limited to ferrofluids, which is a rich and complex system on its own. We believe that a more relevant setup for magnetized plasmas would be a harmonic trap along the external magnetic field.

---

## Round 1 · Referee Report · Anonymous (Referee 3) · 2023-12-12

Strengths

  1. A step towards investigation of paryty breaking ferrofluids

Weaknesses

  1. The Poisson bracket formalism is not the most efficent tool to study hydrodynamics with dissipation.
  2. Strong overlap with previous results.

Report

This paper delves into the unique properties of ferrofluids, synthetic magnetic colloids composed of magnetized nanoparticles each surrounded by a repulsive surfactant layer. When exposed to an external magnetic field, ferrofluids acquire a macroscopic magnetization density, resulting in magnetic behavior deeply intertwined with the fluid dynamics of the surrounding medium. The study draws parallels between ferrofluids and chiral active fluids, specifically those containing unidirectionally spinning hematite cubes, which exhibit a two-dimensional non-dissipative odd viscosity term. However, unlike these chiral fluids, standard ferrofluid dynamics typically do not demonstrate 3D parity breaking terms, likely due to the small size of the magnetic particles.
The manuscript aims to explore whether ferrofluids can manifest a unique 3D odd viscosity term through specific mechanisms. The paper posits that coupling the fluid vorticity with the magnetization, especially when magnetization aligns with a uniform and static applied field, introduces parity breaking terms in ferrofluid hydrodynamics. This results in the emergence of a three-dimensional odd viscosity term. The result is interesting; however, I have a couple of critical comments about the construction.
1. The limit I going to zero seems to be a singular limit. At the level of the discussion presented in the paper it is not clear that this limit is always consistent throughout the manuscript. In the Hamiltonian in this limit the terms involving magnetization are negligible.
2. The internal degrees of freedom of particles usually become unimportant in the hydrodynamic regime as they lead to gapped modes that relax quickly. One can imagine that the authors want to include them, but this leads to a framework that goes beyond pure fluid dynamics and this does not seem to be the message of the paper.
3. Finally, I think that the non-Newtonian aspects of the ferrofluids should also be important for the Helle-Shaw solution and may modify the prediction. In the current form the solution seems to be very similar to Ref. [32].
The authors should address the above points.
  • validity: ok
  • significance: ok
  • originality: low
  • clarity: good
  • formatting: good
  • grammar: excellent

Author:  Dylan Reynolds  on 2024-07-18  [id 4636]

(in reply to Report 3 on 2023-12-12)

Response to Referee 3

Strengths:

  1. A step towards investigation of parity breaking ferrofluids

Weaknesses:

  1. The Poisson bracket formalism is not the most efficient tool to study hydrodynamics with dissipation.

DR, GMM, and SG: Indeed, the Poisson bracket formalism relies on the existence of a Hamiltonian system and it alone cannot capture dissipative effects. However, it can serve as the baseline in which dissipative terms can be added. It is used to arrive at coarse-grained hydrodynamic descriptions (P. C. Hohenberg and B. I. Halperin, Rev. Mod. Phys. 49, 435 (1977); H. Stark and T. C. Lubensky, Phys. Rev. E 67, 061709 (2003); and P. M. Chaikin and L. C. Lubensky, Principles of Condensed Matter Physics, Cambridge University Press, 1995), where dissipative effects can be introduced through a Rayleigh dissipation function. Another advantage of using the Poisson algebra as the starting point is that most parity-odd terms are non-dissipative, which makes the Hamiltonian framework a natural starting point to study them. In fact, the introduction of a new coupling between vorticity and magnetization is most clearly described using a Hamiltonian and Poisson brackets formalism, enabling us to track their microscopic origin. Furthermore, this coupling can be universally applied to any magnetized fluid, even dissipationless ones.

  1. Strong overlap with previous results.

DR, GMM, and SG: Ferrofluids possess many features that break parity at the microscopic scale, but it is puzzling why, at the macroscopic scale, most observables seem to show no sign of parity breaking. Motivated by this puzzle, the main question we are addressing in this manuscript is: what are the conditions for a ferrofluid to exhibit the parity-odd phenomena studied in Ref. [2]? As we showed in this manuscript, the coupling term introduced in Ref. [24] does not generate any parity-odd effect on ferrofluids, since they are washed out by the small size of the ferrofluid particles. The terms ℓiωi and Miωi do share the same mathematical structure, but they correspond to distinct microscopic systems and different ferrohydrodynamic equations. Moreover, if observed, the presence of parity-breaking effects in ferrofluids begs the question of the microscopic origin of such coupling. Ref. [2] introduces the investigation of parity-breaking terms in 3D fluids within the Hele-Shaw framework. This work inspired us to explore this path, as ferrofluids are the most commonly studied fluids in the Hele-Shaw setup. Our present study investigates how parity breaking could potentially arise in ferrofluids, a topic not previously addressed in ferrofluids literature. Furthermore, it serves to provide an extremely simple “tabletop” experiment that can measure 3D odd viscosity. The existing chiral active experiments are limited to the 2D odd viscosity (Ref. [1]), but, our setup allows us to probe the 3D odd viscosity in a simple Hele-Shaw setup. Ref. [1] setup consists of large tanks of fluid subject to rotating magnets. Both ferrofluids and Hele-Shaw cells are readily purchased online, and with a large enough applied magnetic field, our results can be verified. The second novel feature of this work is the possibility that microscopic properties of the ferrofluid may be readily measured via macroscopic properties.

Report: This paper delves into the unique properties of ferrofluids, synthetic magnetic colloids composed of magnetized nanoparticles each surrounded by a repulsive surfactant layer. When exposed to an external magnetic field, ferrofluids acquire a macroscopic magnetization density, resulting in magnetic behavior deeply intertwined with the fluid dynamics of the surrounding medium. The study draws parallels between ferrofluids and chiral active fluids, specifically those containing unidirectionally spinning hematite cubes, which exhibit a two-dimensional non-dissipative odd viscosity term. However, unlike these chiral fluids, standard ferrofluid dynamics typically do not demonstrate 3D parity breaking terms, likely due to the small size of the magnetic particles. The manuscript aims to explore whether ferrofluids can manifest a unique 3D odd viscosity term through specific mechanisms. The paper posits that coupling the fluid vorticity with the magnetization, especially when magnetization aligns with a uniform and static applied field, introduces parity-breaking terms in ferrofluid hydrodynamics. This results in the emergence of a three-dimensional odd viscosity term. The result is interesting; however, I have a couple of critical comments about the construction.

  1. The limit I going to zero seems to be a singular limit. At the level of the discussion presented in the paper it is not clear that this limit is always consistent throughout the manuscript. In the Hamiltonian in this limit the terms involving magnetization are negligible.

DR, GMM, and SG: The I → 0 limit is a standard procedure in the ferrofluids analysis. It analogous to taking the mass m → 0 in the kinetic energy written as p^2/2m. One needs to remember that the square of the p = mv term in the numerator goes to zero faster and hence this is not a singular limit. Similarly, we must keep in mind that the angular momentum density is related to the particle rotation rate via ℓi = IρΩi, and so I → 0 also means that ℓi → 0. In the Hamiltonian, then, the limit must be taken carefully. This could be avoided by writing the Hamiltonian in terms of Ωi or taking this limit in the equations of motion.

  1. The internal degrees of freedom of particles usually become unimportant in the hydrodynamic regime as they lead to gapped modes that relax quickly. One can imagine that the authors want to include them, but this leads to a framework that goes beyond pure fluid dynamics and this does not seem to be the message of the paper.

DR, GMM, and SG: It is not entirely clear to us what the referee is suggesting here. This is exactly what we have done in this paper. The intrinsic angular momentum of the ferrofluid particles relaxes almost instantaneously to follow the fluid vorticity and the magnetization field aligns to M 0. Upon their relaxation, the only degree of freedom left in the system is the fluid velocity.

  1. Finally, I think that the non-Newtonian aspects of the ferrofluids should also be important for the Hele-Shaw solution and may modify the prediction. In the current form the solution seems to be very similar to Ref. [2].

DR, GMM, and SG: Ref. [2] introduces the investigation of parity-breaking terms in 3D fluids within the Hele-Shaw frame- work. This work inspired us to explore this path, as ferrofluids are the most commonly studied fluids in the Hele-Shaw setup. Our present study investigates whether and how parity breaking manifests in ferrofluids, a topic not previously addressed in ferrofluid literature. If evidence of parity breaking is found in ferrofluids, they could provide a simple ‘tabletop’ experiment within a Hele-Shaw cell, avoiding the need for complex or active fluids. The non-Newtonian features of ferrofluids, such as friction reduction may alter some details of the calculation but we do not expect them to change the key qualitative results of the far field circulation. Once again we reiterate that Ref [2] is not about ferrofluids as when we wrote that paper we were puzzled by the fact that the ferrofluids despite possessing all the parity-breaking ingredients at the microscopic scale do not do it at the macro scale. The authors should address the above points.

---

## Round 2 · Referee Report · Anonymous (Referee 1) · 2024-7-29

Report

The resubmitted manuscript addresses all my previous questions and comments satisfactorily. The paper is well-written and presents new and interesting physics. I recommend publishing it in SciPost Physics in its current form.

Recommendation

Publish (surpasses expectations and criteria for this Journal; among top 10%)

---

## Round 2 · Referee Report · Anonymous (Referee 2) · 2024-8-3

Report

I do not understand reference [47] which tells the reader to contact Ferrotec for data on the hydro properties of ferrofluids. If this is straightforward then the author could have contacted Ferrotec, if it isn't then why give the reader the runaround? In all the articles and books cited in ferrohydrodynamics, did nobody ever bother to mention any numbers? If I check [34], I find that the first sentences discuss data on hydro properties of ferrofluids.

In the rebuttal reference [33] is mentioned which confirms the incompressible nature of ferrofluids. I do not understand why this is only stated in the rebuttal and not in the revised manuscript; this is such a central assumption in all of the computations and it is absolutely not obvious that this holds.

I sympathize with the idea that as a theorist one works in a more exploratory way where one is not necessarily bothered too much with the experimental realizability and that when asked about details on experimental realizability one replies with "we leave details of ferrofluid experiments to future work", however the nature of this work is such that it is composed of theoretical insights from previous works which are put together in a modified form with the claim that a very specific system can potentially confirm the corresponding theoretical results. In this case, I believe it is justified to ask this question as I believe this to be the bottom line.

Section IV is new and is seemingly added in part to address my comments about not discussing how the parity-breaking term in (34) can be understood as an entry in the viscosity tensor. However, I think some statements in relation to time-reversal symmetry are incorrect or at least very misleading. It is implied in this section that shear viscosity breaks time-reversal symmetry, just like odd viscosity does. This is not correct, it is only odd under T, which it should be since it is a diagonal dissipative coefficient. The fact that it is odd means it is covariant under time-reversal. Shear viscosity occurs for fluids that are microscopically T-symmetric, whereas odd viscosity requires intrinsic T-breaking, meaning that P-breaking fluids with PT-symmetry display both odd viscosity and shear viscosity. It cannot display terms that are truly PT-breaking, although in this specific case it is not possible to write down such a term. To properly constrain these terms requires accounting for the Onsager-Casimir relations.

I also think section IV is somewhat excessive, since the point of the preceding sections is to show how a specific odd entry in the viscosity tensor arises so why do we now need this general analysis based on symmetry and the second law. I also do not understand why after (38) shear viscosity is seemingly discarded ("So, instead of focusing on a general viscosity tensor, we will only focus on PT -symmetric terms") but then it is brought back again in section VI. It seems that this section builds towards the conclusion "Therefore, in this case, referring to parity-odd viscosity coefficients by 3D odd viscosity is only accurate upon the identification νo⊥ = −2νo∥." which is the same as saying that not all the odd viscosity coefficients allowed by symmetry are independent.

In summary, I think section IV is completely out of place in the manuscript. It saddens me that this is so, because as mentioned I specifically requested a discussion of the parity-breaking term as an entry in the viscosity tensor but this is definitely not what I had in mind.

Requested changes

  • It says "in the in the"
  • Remove section IV, but just explain how this parity-breaking entry in the momentum balance equation is a specific type of 3d odd viscosity -Provide more details on ferrofluids

Recommendation

Ask for major revision

---

## Round 2 · Referee Report · Anonymous (Referee 3) · 2024-9-11

Report

I thank the authors for their careful responses. However, there are still some points in the revised version that remain unclear:

  1. The authors mention that the term "odd viscosity" is used broadly. Given the current logic of the manuscript, is the condition that the two parity-breaking odd coefficients in 3D are related because the authors aim to connect with 2D odd viscosity, or because they assume the system is Hamiltonian? Additionally, the authors invoke relativistic plasmas for some reason. In both relativistic and non-relativistic plasmas, the relation in the collisionless limit holds as computed from kinetic theory. However, in both cases, collisions break this relation. I do not see a clear argument for why such a relation should hold in ferrofluids. This seems to relate to the previous criticism about whether describing hydrodynamics using Hamiltonians is a generic approach or not. The current discussion still suggests that the formalism imposes some constraints.

Another way to approach the problem is through kinetic theory. One could argue that Hall viscosities are non-dissipative, allowing us to consider equilibrium distributions to understand them. However, this doesn't work because, despite being non-dissipative, the coefficients are in front of gradients, meaning we need to move away from equilibrium to observe them. Somehow, the authors appear to move in the opposite direction by starting from a collisionless limit and drawing general conclusions about odd viscosities. I don’t see how this approach is correct. Specifically, if one uses different methods to arrive at hydrodynamic equations and the magnetization-vorticity coupling proposed in the paper, would the result be the same?

  1. To clarify my question about magnetization: is the proposed transport coefficient for ferrofluids only non-zero in the quasi-hydrodynamic regime, where magnetization has not yet relaxed, or in the purely hydrodynamic regime? I don’t see how a field that has relaxed or been integrated out can still influence the transport coefficients. It would be beneficial to see more details included

Additionally, a minor question: is $M^0$ the same as $| \vec{M}^0| $?

Recommendation

Ask for major revision

---

## Round 2 · Author Response

Dear Editor,

We are resubmitting our manuscript entitled "Parity Breaking in Ferrofluids with Vorticity-Magnetization Coupling" for your consideration for publication in SciPost. We sincerely thank the referees for their time and effort in reviewing this paper. Referee 1 provided a positive review and requested additional discussion on the microscopic origins of the new parity-breaking term. In response, we have expanded the discussion to include more microscopic arguments. We have also put this coupling into context by comparing it with that of Ref. [24].

Referees 2 and 3 raised concerns about the novelty of this work and its comparisons with Ref. [24] by Markovich and Lubensky, as well as our previous work on Hele-Shaw cells and parity breaking fluids. We believe there is some misunderstanding, which we have addressed in the referee reports and reiterate below. Ferrofluids possess many features that break parity at the microscopic scale, but it is puzzling why, at the macroscopic scale, most observables seem to show no sign of parity breaking. To the best of our knowledge despite vast literature on the ferrofluids, there has been no discussion on the parity breaking in them. Motivated by this puzzle, our main question in this manuscript is: what are the conditions for a ferrofluid to exhibit parity breaking phenomena in a Hele-Shaw experiment? Our past work on Hele-Shaw flows motivated this work and raised whether Ferrofluids can break parity. One of the main reasons we highlighted Ref. [24] in our work is to demonstrate that following this work does not lead to any parity breaking in ferrofluids due to the small size of the magnetic particles. This is one of the results presented in our manuscript. Ref. [24] is more applicable to ferrofluids with bigger colloidal particles like those used in chiral active fluids (Ref. [1]). The novelty of our work is the new mechanism that can break parity in ferrofluids by coupling magnetization to ambient vorticity. In the end, we provide a simple experimental test for this prediction that can confirm or refute our predictions.

Finally, based on the points above, we strongly believe that our work should be considered for SciPost Physics, but in the interest of the time already spent, we would accept publishing it in SciPost CORE, if the editor still believes this to be the best fit for our current work. We will abide by the editorial decision on this matter.

Yours sincerely,

D. Reynolds, G. M. Monteiro, S. Ganeshan

---

## Round 2 · List of Changes

• Expanded the heuristic argument for the new coupling with a more in depth analysis of the microscopics.
• More clearly highlighted the aspects of our analysis which are unique to ferrofluids, giving a stronger connection to odd viscosity. This is done by splitting section 3 into two parts.
• Changed the title of the paper to “Parity breaking in….” instead of “3D Odd Viscosity in…”
• Changed brackets and Hamiltonian from velocity to momentum density, to garuntee the brackets satisfy the Jacobi identity. The incompressible limit is taken at the level of the equations of motion.
• Changed reference in abstract to a numbered citation.
• Restructured the third paragraph of the introduction to avoid mention of 3D active matter and more clearly state what we investigate.
• Added a new section reviewing the anisotropic viscosity tensor in 3D.
• Clearly indicated (with italics) the main results of each section, drawing attention to what is required for parity breaking in ferrofluids.
• Grammatical issues and typos in some inline equations.
• Formatting

---

## Editorial Decision

awaiting_resubmission